# Presenilins and γ-Secretase in Membrane Proteostasis

**DOI:** 10.3390/cells8030209

**Published:** 2019-03-01

**Authors:** Naoto Oikawa, Jochen Walter

**Affiliations:** Department of Neurology, University of Bonn, 53127 Bonn, Germany; Naoto.Oikawa@ukbonn.de

**Keywords:** presenilin, γ-secretase, autophagy, Alzheimer disease, proteostasis, intramembrane proteolysis, membrane trafficking

## Abstract

The presenilin (PS) proteins exert a crucial role in the pathogenesis of Alzheimer disease (AD) by mediating the intramembranous cleavage of amyloid precursor protein (APP) and the generation of amyloid β-protein (Aβ). The two homologous proteins PS1 and PS2 represent the catalytic subunits of distinct γ-secretase complexes that mediate a variety of cellular processes, including membrane protein metabolism, signal transduction, and cell differentiation. While the intramembrane cleavage of select proteins by γ-secretase is critical in the regulation of intracellular signaling pathways, the plethora of identified protein substrates could also indicate an important role of these enzyme complexes in membrane protein homeostasis. In line with this notion, PS proteins and/or γ-secretase has also been implicated in autophagy, a fundamental process for the maintenance of cellular functions and homeostasis. Dysfunction in the clearance of proteins in the lysosome and during autophagy has been shown to contribute to neurodegeneration. This review summarizes the recent knowledge about the role of PS proteins and γ-secretase in membrane protein metabolism and trafficking, and the functional relation to lysosomal activity and autophagy.

## 1. Introduction

Since mutations causative of early-onset familial Alzheimer disease (FAD) have been identified in the *PSEN1* and *PSEN2* genes [1,2,3], pathophysiological functions of the encoded proteins presenilin 1 (PS1) and PS2 have been extensively studied. PS1 and 2 are multifunctional proteins involved in fundamental cellular events, such as cell differentiation, intracellular signaling, and membrane trafficking [4]. The best-characterized function of PS proteins is their role in intramembrane proteolysis as the catalytic component of the γ-secretase complex [5,6,7]. A fundamental role of γ-secretase in cell differentiation and development has been attributed to the regulated intramembrane cleavage of Notch, and *PSEN1*/*PSEN2* double knockout (PSsDKO) mice that completely lack γ-secretase activity closely resemble the developmental phenotypes of Notch-deficient mice [8,9]. PS proteins also exert critical roles in the maintenance of cellular homeostasis and function by modulating membrane protein degradation [10] and intracellular vesicle/protein trafficking [4,11].

Lysosomal activity and autophagy are critical for cellular function and viability by maintaining cellular homeostasis of lipids, sugars, and proteins [12]. Impairment of metabolite degradation in lysosomes and during autophagy could promote neurodegeneration and, thus, could also be implicated in the pathogenesis of AD [13,14,15]. Noticeably, it has been reported that FAD-related clinical PS mutants could impair autophagy and lysosomal activity, and disturb membrane protein metabolism. 

This review provides an overview of the present knowledge on the role of PS proteins in the homeostasis of membrane protein metabolism that might be dependent or independent on the catalytic activity of γ-secretase.

## 2. Presenilins

In humans, there are two homologous genes, *PSEN1* and *PSEN2*, located on chromosome 14 and 1, respectively. The encoded proteins PS1 and PS2 are multi-pass transmembrane proteins synthesized into the membrane system of the endoplasmic reticulum (ER) [16,17,18]. After synthesis and during transport from the ER to Golgi compartments, full-length PSs undergo endoproteolysis and assembly with the additional proteins nicastrin (NCT), anterior-pharynx defective-1 (APH1), and presenilin-enhancer-2 (PEN2) to form catalytically active γ-secretase complexes [19,20,21,22,23,24,25,26,27,28,29] (Figure 1). Besides the localization in the ER and Golgi [30], PS proteins are also found in additional subcellular membrane systems, including the plasma membrane (PM) [31], endosomes [32], lysosomes [33], nuclear envelope [34], and mitochondria [35]. However, while PS1 mainly localizes in the Golgi compartment, at the PM, and in endosomes, PS2 is predominantly localized in endosomes and lysosomes [36,37]. The localization of PS2 in the endolysosomal compartment depends on the phosphorylation at serine residue 19 in its intracellular N-terminal domain [37]. PS2 can be phosphorylated at additional sites in the N-terminal domain and within the hydrophilic loop domain by several protein kinases, including aurora A, casein kinase-1 (CK1) and CK2 [17,37,38,39]. PS1 can also be phosphorylated within the hydrophilic loop domain at several sites upon generation of C-terminal fragments (CTFs) by endoproteolytic cleavage [20,38,40,41,42,43,44,45]. However, the effects of phosphorylation in the CTFs on the activity and localization of γ-secretase are not clear. Rather, the phosphorylation of PS1 and PS2 CTFs inhibits their degradation by caspases, retards the progression of apoptosis, and regulates the interaction with β-catenin and related signaling pathways [41,42,43,44]. The phosphorylation of PS1-CTFs also affects autophagy (see below) [46,47]. PSs can also undergo ubiquitination in vitro [48], but its effect on proteasomal degradation of PSs is not clear. Due to efficient endoproteolysis, the cellular level of full-length PSs (PSs-FL) is very low, while substantial amounts of PSs-N-terminal fragments (NTFs) and -CTFs are detected [19,49,50,51,52]. PSs are expressed not only in neurons and glial cells [53,54,55,56,57,58,59] but also in tissues outside the brain [8].

## 3. Functions of Presenilins in Membrane Protein Metabolism

### 3.1. γ-Secretase as an Intramembrane Cleaving Protease

The most extensively characterized function of PS proteins is that as the catalytic components of γ-secretase complexes in assembly with additional proteins NCT, APH1, and PEN2 [22,23,24] (Figure 1). γ-Secretase is an aspartyl protease responsible for proteolysis of type I membrane proteins [5,6,7]. It has been proposed that the substrate proteins—which generally have short ectodomains [60,61,62] and are recognized by γ-secretase, possibly by NCT [63,64,65,66]—initially bind to a substrate-docking site, and are subsequently passed to the active site of γ-secretase where the α-helical conformation of the substrate’s transmembrane domain is focally changed to a β-sheet conformation to enable hydrolytic cleavage [67,68]. Details on the molecular features of γ-secretase and substrate processing have been described in recent excellent reviews [5,69,70].

More than 90 membrane proteins have been identified as γ-secretase substrates [71,72]. An important characteristic is that the protease usually does not cleave substrates with large ectodomains, although cleavage of full-length amyloid-precursor-like protein 1 (APLP1) and E-Cadherin has been described [31,73]. Rather, γ-secretase-dependent cleavage occurs after precedent shedding of the respective ectodomains of the individual substrate proteins. Ectodomain shedding of type I membrane proteins can be exerted by many different proteases, including members of the A Disintegrin And Metalloproteinase (ADAM) family, many additional metalloproteases, and several aspartyl-, serine, and cysteine proteases [74]. Since shedding usually occurs close to the transmembrane domain, the remaining membrane-tethered CTFs have relatively short ectodomains. It has been shown that the length of the ectodomain can affect the efficacy of the subsequent cleavage of these CTFs by γ-secretase [60,61,62]. Thus, γ-secretase cleavage usually generates two different products: short extracellular fragments that can be secreted from cells and intracellular domains that are liberated into the cytosol. While it is reported that the extracellular fragment of a γ-secretase substrate, the B-cell maturation antigen (BCMA), mediates survival of activated B cells [62], fundamental physiological functions of extracellular fragments of other substrate proteins of γ-secretase remain to be established. On the other hand, the intracellular domains of select substrates can regulate signal transduction pathways and gene transcription. The best-understood example is the intracellular domain (ICD) of Notch that, when liberated from cellular membranes by γ-secretase cleavage, can translocate to the nucleus to regulate transcription and determine cell fate decisions [75,76]. Nuclear translocation, control of gene transcription, and modulation of intracellular signaling pathways have also been shown for the ICDs of the amyloid precursor protein (APP) [77,78], ErbB4 [79,80,81,82,83,84,85], CD44 [86,87,88,89,90,91,92], and ephrinB2 [93,94,95]. However, for most of the γ-secretase substrates, a physiological role for the cleavage products has not been identified, and it has been speculated that γ-secretase activity might also be important for the degradation of CTFs generated by ectodomain shedding that otherwise would remain in cellular membranes and potentially interfere with membrane dynamics [60,96].

Notably, the conditional deletion of PS proteins or other components of the γ-secretase complex in neurons causes age-dependent neurodegeneration, and it has been speculated that this phenotype could involve the accumulation of membrane-tethered γ-secretase protein substrates [97,98,99]. Indeed, transgenic overexpression of APP CTFs in mouse brain neurons impaired synaptic function and caused neurodegeneration [100,101]. Evidence from cellular and mouse models indicates that increased levels of APP CTFs could interfere with mitochondrial function by changing membrane contacts with the ER [102], endocytosis [103,104], axonal trafficking of neurotrophic factors [105,106], neurite outgrowth [107], and synaptic plasticity [108]. While the exact mechanisms underlying these effects remain to be further characterized, it has been proposed that accumulated APP CTFs might interfere with intracellular signaling and subcellular trafficking by increased interaction with endocytic adaptor proteins, FE65 and ARH [109], heterotrimeric G_o_-proteins [107], or the Ca^2+^ sensor synaptotagmin 7 [108]. Impaired signaling upon inhibition of γ-secretase has also been observed for the triggering receptor expressed on myeloid cells 2 (TREM2) [110,111]. This immune receptor mediates signaling in macrophages and microglia via its co-receptor Dap12 [112,113]. Since accumulated TREM2 CTFs can also interact with Dap12, it was proposed that the TREM2 CTF lacking the ligand binding ectodomain upon ectodomain shedding competes with functional full-length TREM2 for coupling to limited amounts of Dap12.

In the future, it will be important to further understand the involvement of substrate accumulation in cellular dysfunction observed upon genetic and pharmacological inhibition of γ-secretase in different cell types. Here, it is also interesting to note that certain mutations in the genes encoding PS1 or the γ-secretase components NCT and PEN2 are associated with hidradenitis suppurativa [114,115]. Although impaired cleavage of Notch might be implicated [116], additional mechanisms could also contribute to pathogenesis [117].

### 3.2. Presenilins as Modulators of Vesicle/Protein Trafficking

In addition to the catalytic function within the γ-secretase complex, PS proteins also interact with several proteins to modulate their biological activity, such as β-catenin and glycogen synthase kinase 3β (GSK3β) for Wnt/β-catenin signaling and inositol 1,4,5-trisphosphate receptor (InsP_3_R), ryanodine receptors, and sarco/endoplasmic reticulum Ca^2+^-ATPase (SERCA) pump for cellular calcium homeostasis [71]. Presenilins also interact with proteins involved in intracellular vesicular trafficking, including Rab11 [118], Rab guanosine diphosphate dissociation inhibitor (RabGDI) [119], phospholipase D1 (PLD1) [120], syntaxin 1A [121], syntaxin 5 [122], X11α/β [123], and Annexin A2 (featured in the next paragraph) [47]. Interestingly, PS1 deficiency accelerates while FAD-linked PS1 mutants impair anterograde transport of APP-containing vesicles [124,125]. Additionally, PS1 deficiency negatively affects bulk or receptor-mediated endocytosis and transcytosis [57,109,126,127,128]. In line with a role of PS in membrane trafficking, the localization and transport of several membrane proteins, including APP [124,129,130], APLP1 [129], C-terminal fragment of APP (APP-CTF) [131,132], TrkB [129], N-Cadherin [133], intracellular adhesion molecule 5 (ICAM5) [134], NMDA (N-methyl-D-aspartate) receptor [135], transferrin receptor [136], tyrosinase [137], epidermal growth factor receptor (EGFR) [138], integrin β1 [139], low-density lipoprotein (LDL) receptor [109], vATPase V0a1 subunit [140], EphB [141], LDL receptor-related protein 1 (LRP1) [127], and the triggering receptor expressed on myeloid cells 2 (TREM2), are altered in PS-deficient or PS-mutant cells [128]. Interestingly, PS-dependent vATPase V0a1 transport and maturation can have an impact on lysosomal acidification and subsequent lysosomal/autophagic function (see below).

## 4. Presenilin in Autophagy

Autophagy is important for neural viability, indicated by the age-dependent neurodegeneration and intraneuronal accumulation of protein aggregates observed in mouse brains upon conditional deletion of key autophagy genes, ATG5 or ATG7, in neuronal progenitor cells [142,143]. Autophagy also contributes to neurogenesis and differentiation [144], neuronal maturation [145], and synaptic refinement [146]. Due to the importance of autophagy in the maintenance of neural cells, impaired autophagy has been suspected to contribute to the development of neurodegenerative diseases, such as Parkinson disease, Huntington disease, and Alzheimer disease [13,14,15]. 

Autophagy describes a process in which cellular material is targeted to the lysosome/vacuole for degradation and recycling of macromolecular constituents [147]. Three types of autophagy have been described in mammals: macroautophagy, chaperone-mediated autophagy, and microautophagy. Macroautophagy (hereafter called autophagy) involves several steps: phagophore formation by vesicle nucleation, vesicle elongation and sequestration of cytoplasmic material, autophagosome formation by phagophore maturation, autolysosome formation by fusion of autophagosome and lysosome, and eventually degradation of autolysosomal contents [12,14]. Each of these steps is coordinated by a panel of proteins [12,14].

The first indication on a relation of PS proteins and autophagy resulted from the observation that *PSEN1*-null mouse fibroblasts and neurons showed enlarged lysosomal organelles with an accumulation of α- and β-synucleins [148]. The accumulation of synucleins appeared to be caused by increased production and decreased degradation rather than from impairment of axonal transport and could be rescued by the expression of wild-type or clinical mutant PS1 or by modulation of cellular calcium homeostasis. Enlarged lysosomal organelles and mislocalization of synucleins could not be induced by pharmacological inhibition of γ-secretase in PS1-expressing neurons. Essenlens et al. also reported a possible defect of autophagy by showing an accumulation of telencephalin (ICAM5) in autophagic vacuoles in *PSEN1*-null mouse neurons [134]. The increase in ICAM5 appeared to result from decreased degradation, not from the impaired transport of newly synthesized ICAM5 in the early secretory pathway. Accumulation of ICAM5 was detected exclusively in autophagic vacuoles, which are not acidified in *PSEN1*-null neurons, suggesting impairment in the fusion of autophagosomes with lysosomes. Noticeably, ICAM5 accumulation in *PSEN1*-null cells could be rescued by the expression of wild-type or dysfunctional PS1, but not by pharmacological inhibition of γ-secretase in wild-type neurons. Cataldo et al. also reported an increase of lysosomal proteins cathepsin D (CatD) and the cation-independent 215-kDa-form mannose-6-phosphate receptor (MPR215), which targets lysosomal proteins to lysosomes, in human and mouse brains harboring clinical *PSEN1* or *PSEN2* mutations [149]. Together, these studies indicate the implication of PS1 and/or PS2 in lysosomal function and autophagy, probably independent of the catalytic activity of γ-secretase.

In line with this notion, Lee et al. reported decreased turnover of long-lived proteins in *PSEN1*- KO blastocysts associated with increased formation of autophagosomes [140], suggesting impaired autophagy. Ultrastructural analysis revealed the accumulation of enlarged autophagosomes and early autolysosomes containing undigested materials, suggesting impaired clearance of autophagic vacuoles after fusion with lysosomes. In support of dysfunctional vesicle fusion and lysosomal impairment, *PSEN1*-KO cells showed decreased maturation of CatD and increased lysosomal pH. However, the impairments in lysosomal acidification, cathepsin B (CatB) activity, and CatD processing were not mimicked by pharmacological or genetic inhibition of γ-secretase activity. Thus, the acidification defect in *PSEN1*-KO cells occurred independently of the catalytic activity of γ-secretase and was attributed to impaired interaction of v-ATPase V0a1 subunit with full-length PS1 that resulted in reduced maturation and targeting to the lysosome. Defective autophagosome accumulation and acidification were also observed in brains of PS1-deficient mice, as well as in human fibroblasts with FAD mutant PS1, in which impaired autophagy could be rescued by restoring lysosomal pH using cyclic adenosine monophosphate (cAMP) [150]. Neely et al. also reported impaired proteolysis of long-lived protein together with alterations in other markers of autophagy in PSsDKO and FAD mutant cells [151]. However, the acidification of lysosomes was not impaired, but rather enhanced in PSsDKO cells. Noticeably, pharmacological inhibition of γ-secretase activity did not induce similar phenotypes, while the expression of catalytically inactive or FAD mutant PS1 in *PSEN1*-null cells normalized levels of the microtubule-associated protein 1 light chain 3 (LC3) II that accumulated in PS-deficient cells. 

Together, these studies indicate that alterations in the endolysosomal system and autophagy in PS-deficient cells could be independent of the catalytic activity of γ-secretase, and suggest a role of PS proteins in the fusion of autophagosomes with lysosomes. However, there is controversy in the field regarding the molecular mechanism by which PS regulates autophagy. For example, Zhang et al. did not observe an impairment of autophagic flux, cellular vesicular acidification, CatD maturation and expression, and v-ATPase V0a1 subunit glycosylation with showing no physical interaction with full-length PS1 [152]. Instead, the PS-dependent regulation of the CLEAR (coordinated lysosomal expression and regulation) gene network encoding components for lysosomal biogenesis was revealed. Coen et al. also reported normal lysosomal acidification but a significant alteration in lysosomal calcium storage/release in PSsDKO cells [153]. Decreased lysosomal calcium levels were found in PSsDKO mouse embryonic fibroblasts (MEFs) and *PSEN1*-KO neurons; noticeably, this defect was rescued by expression of wild-type or γ-secretase-dysfunctional PS1 in PSsDKO MEFs. Neely Kayala et al. also reported lower lysosomal calcium levels in PSsDKO MEFs, and altered expression of lysosomal calcium efflux channels, known as two-pore channels (TPC). These alterations were not only observed in PSsDKO, but also in *PSEN1* and *PSEN2* single-KO MEFs [154]. Decreased lysosomal Ca^2+^ levels were also observed in *PSEN1*-KO blastocysts, and was rescued by inhibition of the endolysosomal transient receptor potential cation channel mucolipin subfamily member 1 (TRPML1), indicating the involvement of TRPML1 in the PS-dependent regulation of lysosomal calcium [155]. Interestingly, impaired autophagy in the *PSEN1*-KO blastocysts could be restored by pharmacological normalization of lysosomal pH but not by modulation of lysosomal calcium level. Additionally, the decrease in the lysosomal calcium level could be induced by pharmacological inhibition of vATPase and a subsequent increase in lysosomal pH, suggesting that the impairment in lysosomal acidification in PS-deficient cells results in a disturbance of lysosomal calcium homeostasis and autophagy.

Impairment in autophagy in PS mutant cells has also been attributed to altered expression and activity of acid sphingomyelinase (ASM) [156]. ASM levels in blood plasma and in fibroblasts from individuals carrying FAD-associated *PSEN1* mutations were increased together with an accumulation of LC3II and long-lived proteins under serum-starved conditions. Interestingly, the increase of LC3II could be induced by the addition of ASM to wild-type fibroblasts and neurons.

Another PS-dependent mechanism related to the regulation of autophagy could involve the mechanistic target of rapamycin complex 1 (mTORC1) and transcription factor EB (TFEB) [157]. PSsDKO and *PSEN1*-KO MEFs showed dysregulated lysosomal amino acid sensing by TORC1, which inhibits the activation and translocation of TFEB to the nucleus and subsequent CLEAR gene network expression. Noticeably, the impairment of CLEAR gene network activity could be rescued by the expression of wild-type PS1, PS2, or the γ-secretase-dysfunctional form of PS1, but not by the expression of clinical PS1 mutants. Additionally, pharmacological γ-secretase inhibition did not impair CLEAR gene network activity, suggesting that the impairment is independent of γ-secretase activity. It has been proposed that PS-dependent alterations in mTORC1 and CLEAR gene expression involve the calcium/calmodulin-dependent protein kinase (CaMK)-cAMP response element binding protein (CREB) and related sestrin pathway. Impairment of CLEAR gene network activity has also been observed in induced pluripotent stem cell (iPSC)-derived human neurons carrying clinical *PSEN1* mutations, and was associated with increased intracellular amyloid β-protein (Aβ), phosphorylated tau, cleaved caspase 3, and degenerated microtubules. Involvement of TFEB and CREB-mediated nuclear signaling pathway in autophagy impairment have also been observed in other iPSC-derived human neural cells carrying clinical *PSEN1* mutation or lacking *PSEN1* [158,159]. Regarding the CREB-mediated nuclear signaling pathway, reduced ERK activity initiates the activation and translocation of GSK3β to the nucleus, which decreases the expression of CREB and autophagy-related genes in *PSEN1*-KO neural stem cells [159].

The effect of PS proteins in autophagy might also depend on their phosphorylation state [46,47]. Inhibition of PS1 phosphorylation at Ser367 by mutagenesis of this site (PS1-S367A) led to increased LC3II and p62 levels in mouse brain and cultured primary neurons [46]. Levels of APP-CTF cleaved by β-secretase (APP-CTFβ) and of Aβ40 and Aβ42 were also increased in the PS1-S367A mouse brain homogenate. These effects were attributed to the phosphorylation-dependent interaction of PS1 with Annexin A2 and N-ethylmaleimide-sensitive factor attachment protein receptor (SNARE) Vamp8, which, in turn, modulates the fusion of autophagosomes with lysosomes [47].

In contrast to most other reports, PS deficiency has also been found to be associated with the activation of autophagy. Száraz et al. reported activation of autophagy upon knockdown of PS1-, but not of PS2, in hepatocytes, as indicated by increased LC3 protein expression and inactivation of mTOR [160]. In the same study, it was suggested that the alteration in autophagy involved calcium homeostasis-related ER stress response. Autophagy activation has also been reported in bone marrow-derived mesenchymal stem cells (BM-MSCs) upon γ-secretase inhibition [161]. The pharmacological inhibition of γ-secretase activity resulted in increased expression of PTEN, which can stimulate autophagy [162], and in decreased phosphorylation of PI3K, Akt, and mTOR, suggesting that the PTEN-PI3K/Akt/mTOR pathway could activate autophagy upon γ-secretase inhibition in BM-MSCs.

## 5. Concluding Remarks

Extensive research has significantly improved our understandings of the role of PS/γ-secretase in membrane protein degradation, lysosomal activity, and in autophagy (Figure 2), but the underlying mechanisms remain controversial. Thus, it will be important to further dissect the molecular mechanisms and pathways related to PS-dependent membrane protein homeostasis, and to understand the physiological and pathophysiological implications not only in the nervous system but also in the periphery. It would be interesting to investigate whether PS1 and PS2 differentially affect membrane protein metabolism since both proteins show differential subcellular distribution [36,37] and could exert individual functions [163,164]. Another important question to clarified in the future is whether and how clinical PS mutants affect membrane protein homeostasis and contribute to AD development. Deletion of PSs or other γ-secretase components in neurons results in age-dependent neurodegeneration [97,98,99,135], suggesting that PS deficiency or clinical PS mutants can contribute to AD development in addition to known alterations on Aβ production. Cell bioengineering technology, such as iPSC generation and genome editing, can provide powerful experimental systems to identify the molecular effects caused by disease-related mutations in more authentic, isogenic human neural cells. Thus, an examination of membrane protein homeostasis in human neural cells carrying FAD-related clinical *PSEN* mutations could provide important clues to more comprehensively understand how *PSEN* mutations cause AD development.

## Figures and Tables

**Figure 1 cells-08-00209-f001:**
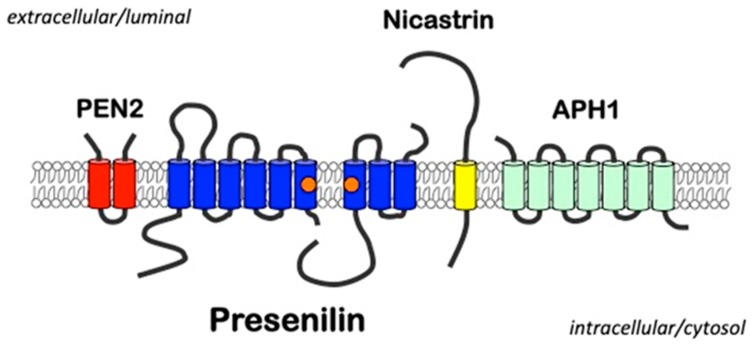
A schematic of the γ-secretase complex. Presenilins represent the catalytic components of γ-secretase complexes that contain three additional proteins presenilin-enhancer-2 (PEN2), nicastrin, and anterior-pharynx defective-1 (APH1). The aspartyl residues in transmembrane domains 6 and 7 required for catalytic activity are indicated by orange circles.

**Figure 2 cells-08-00209-f002:**
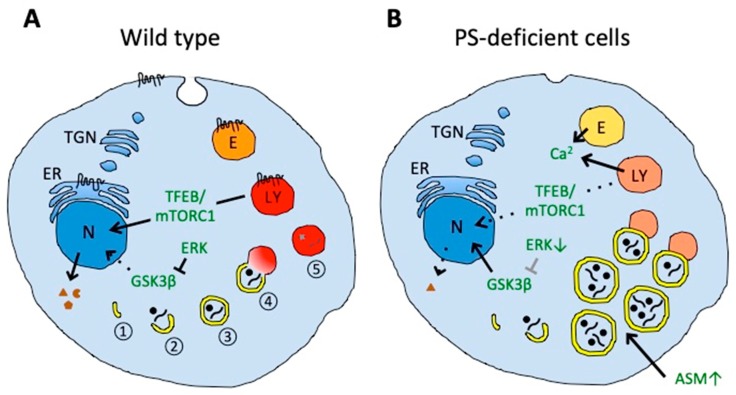
Impaired autophagy and lysosomal degradation in PS-deficient cells. (**A**) In wild-type cells, autophagy includes nucleation (1) and elongation of phagophores (2), autophagosome formation by phagophore maturation (3), autolysosome formation by the fusion of autophagosomes and lysosomes (4), and final degradation of the contents (5). (**B**) In PS-deficient cells, enlarged autophagic vacuoles accumulate and contain undigested engulfed material. The accumulation of autophagic vacuoles could result from the disturbed fusion of autophagosomes with lysosomes, probably caused by impaired lysosomal acidification (illustrated in a pale red color). Aberrant acidification could also affect calcium homeostasis in endolysosomal vesicles that could contribute to impaired vesicle fusion. PS deficiency can also impair amino acid sensing by mTORC1 on lysosomes and decrease activation and nuclear translocation of transcription factor EB (TFEB), thereby decreasing expression of proteins mediating biogenesis of lysosomal and autophagic vesicles. Decreased translocation of glycogen synthase kinase 3β (GSK3β) from the cytosol to the nucleus could also decrease the activation of TFEB. Increased acid sphingomyelinase (ASM) in PS-deficient cells can induce the accumulation of autophagic vacuoles. Decreased endocytosis in PS-deficient cells could also affect membrane protein and lipid homeostasis. Presenilin can be localized in the endoplasmic reticulum (ER), plasma membrane, endosomes (E), and lysosomes (LY). N, nucleus; TGN, trans-Golgi network.

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
