# Peer review of "Presenilins and γ-Secretase in Membrane Proteostasis"

_cells, 2019, doi:10.3390/cells8030209_

Round 1

Reviewer 1 Report

Review for Oikawa and Walter Cells 2019

This review article is focused on the role presenilin/g-secretase has in autophagy and lysosome biology. Because the Alzheimer’s field is filled with reveiwes covering the traditional role of presenilin/g-secretase in the cleavage and accumulation of Abeta, this review offers a refreshing perspective covering different views into the role of presenilin/g-secretase. The authors discuss the primary function of presenilin/g-secretase as an intramembrane protease and its involvement in protein trafficking. A large part of the review does a nice job of covering the primarily literature regarding the role of presenilin/g-secretase in the regulation of autophagy and lysosome function.  

However, despite the nice coverage of autophagy and lysosome function of presenilin/g-secretase, the authors do not cover the role of presenilin/g-secretase in membrane proteostasis in great detail. The title of the article is “Presenilin/g-secretase in membrane proteostasis” and on line 42-43 the authors state “This review provides an overview of the present knowledge on the role of PS/g-secretase in the homeostasis of membrane protein metabolism.” The manuscript only spends a small amount of text discussing this role even though it appears the author wish to make this the center point.  For example the only section of the review that really extends on this topic is on lines 95-99. The authors discuss the many proteins that presenilin/g-secretase may interact with and the influence on their trafficking but homeostasis is not discussed (see lines 110-119). 

Additionally, the review clearly describes the protease independent role of presenilin/g-secretase in regulating autophagy and lysosome biology. Thus the role of presenilin/g-secretase in mediating membrane protein homeostasis seems diffuse. Is protease activity required for the break-down of membrane proteins? Do proteins accumulate in the membrane when presenilin/g-secretase is knocked out or mutated?

A study by Deyts et al., (2016) (eLife 5:e15645), suggests that PS1 has an important role in processing APP to prevent its accumulation in the membrane. This paper may help the authors expand on the idea that presenilin/g-secretase may act to regulate membrane protein levels and homeostasis. 

There are several typos throughout the manuscript. For example, line 9 “ “The two homologues proteins PS1 and PS2 …” should be “The two homologous proteins PS1 and PS2…”; line 17 “…during autophagy has been shown contribute to …” should be “…during autophagy has been shown to contribute to …”; line 107 “(be featured in next paragraph)” should be “(featured in the next paragraph); line 116 “… (TREM2) [92] is…” should be “… (TREM2) [92] are…”; line 123 “…progenitor cells on [107,108]” should be “…progenitor cells [107,108]”; etc.

The sentence on line 180-183 is awkward. 

Figure 1 is not mentioned in the text.

Author Response

1. This review article is focused on the role presenilin/g-secretase has in autophagy and lysosome biology. Because the Alzheimer’s field is filled with reveiwes covering the traditional role of presenilin/g-secretase in the cleavage and accumulation of Abeta, this review offers a refreshing perspective covering different views into the role of presenilin/g-secretase. The authors discuss the primary function of presenilin/g-secretase as an intramembrane protease and its involvement in protein trafficking. A large part of the review does a nice job of covering the primarily literature regarding the role of presenilin/g-secretase in the regulation of autophagy and lysosome function.  

However, despite the nice coverage of autophagy and lysosome function of presenilin/g-secretase, the authors do not cover the role of presenilin/g-secretase in membrane proteostasis in great detail. The title of the article is “Presenilin/g-secretase in membrane proteostasis” and on line 42-43 the authors state “This review provides an overview of the present knowledge on the role of PS/g-secretase in the homeostasis of membrane protein metabolism.” The manuscript only spends a small amount of text discussing this role even though it appears the author wish to make this the center point.  For example the only section of the review that really extends on this topic is on lines 95-99. The authors discuss the many proteins that presenilin/g-secretase may interact with and the influence on their trafficking but homeostasis is not discussed (see lines 110-119). 

Additionally, the review clearly describes the protease independent role of presenilin/g-secretase in regulating autophagy and lysosome biology. Thus the role of presenilin/g-secretase in mediating membrane protein homeostasis seems diffuse. Is protease activity required for the break-down of membrane proteins? Do proteins accumulate in the membrane when presenilin/g-secretase is knocked out or mutated?

Response: We now included a more detail description and discussion of studies that could indicate a role of presenilins and gamma-secretase in the homeostasis of membrane proteins and the potential relation to neurodegenerative processes. This part also includes information about the known functions of cleavage products generated by gamma-secretase mediated cleavage. Please refer to the new section (lines 87-131) in the revised manuscript.  

2. A study by Deyts et al., (2016) (eLife 5:e15645), suggests that PS1 has an important role in processing APP to prevent its accumulation in the membrane. This paper may help the authors expand on the idea that presenilin/g-secretase may act to regulate membrane protein levels and homeostasis. 

Response: The study by Deyts et al., 2016 (reference 106 in the revised manuscript), is now mentioned, together with several additional studies that could indicate a role of accumulated APP C-terminal fragments in cellular (dys)function (new references 100-113).

3. There are several typos throughout the manuscript. For example, line 9 “ “The two homologues proteins PS1 and PS2 …” should be “The two homologous proteins PS1 and PS2…”; line 17 “…during autophagy has been shown contribute to …” should be “…during autophagy has been shown to contribute to …”; line 107 “(be featured in next paragraph)” should be “(featured in the next paragraph); line 116 “… (TREM2) [92] is…” should be “… (TREM2) [92] are…”; line 123 “…progenitor cells on [107,108]” should be “…progenitor cells [107,108]”; etc.

Response: We carefully checked and edited the manuscript.

4. The sentence on line 180-183 is awkward. 

Response: The sentence has been rephrased to “However, there is controversy in the field regarding the molecular mechanism by which PS regulates autophagy

5. Figure 1 is not mentioned in the text.

Response: We now refer to figure 1 in the revised manuscript

We would like to this reviewer for critical reading of the manuscript, very positive comments, and the very helpful comments and suggestions to improve the manuscript.

Reviewer 2 Report

This review article is comprehensive, contemporary and very well-written. There are no corrections I would suggest. Very good job.

Author Response

Thank you very much.

Reviewer 3 Report

In this manuscript, the authors review the literature concerning the membrane protein Presenilin, with emphasis on the role of Presenilin in proteostasis.  This review comes at a necessary time when multiple studies have analyzed the complex relationship between Presenilin and autophagy.

Very minor points, the authors are free to make the proposed changes:

Line 26: “are a multifunctional proteins”, change to “are multifunctional proteins”

Line 78: “g-secretase, at where”, change to “g-secretase, where”

Line 81: “have been described in recent excellent reviews” (5,69-71)

Reference 71 is a report describing the enzyme Quiescin Sulfhydryl Oxidase 2, not gamma-secretase. I suggest eliminating this reference.

Could not find reference 67 referenced in the text, maybe should go in line 79?

Line 151: “catepsin D” change to “cathepsin D”

Line 180. This is an important part of the review, in which the authors express the idea that there is controversy in the field regarding the mechanism by which PS regulates the autophagosome-lysosome fusion. The phrase is not very clear, “ However, the underlying mechanisms are discussed controversially, and remain to be identified in more detail”. I suggest changing to “However, there is controversy in the field regarding the molecular mechanism by which PS regulates autophagy”

I could not find a  reference to Figure 1 in the text.

Author Response

In this manuscript, the authors review the literature concerning the membrane protein Presenilin, with emphasis on the role of Presenilin in proteostasis.  This review comes at a necessary time when multiple studies have analyzed the complex relationship between Presenilin and autophagy.

Very minor points, the authors are free to make the proposed changes:

Line 26: “are a multifunctional proteins”, change to “are multifunctional proteins”

Line 78: “g-secretase, at where”, change to “g-secretase, where”

Line 81: “have been described in recent excellent reviews” (5,69-71)

Reference 71 is a report describing the enzyme Quiescin Sulfhydryl Oxidase 2, not gamma-secretase. I suggest eliminating this reference.

Could not find reference 67 referenced in the text, maybe should go in line 79?

Line 151: “catepsin D” change to “cathepsin D”

Response: We corrected the indicated errors and typos, and carefully checked the manuscript again during revision. 

2. Line 180. This is an important part of the review, in which the authors express the idea that there is controversy in the field regarding the mechanism by which PS regulates the autophagosome-lysosome fusion. The phrase is not very clear, “ However, the underlying mechanisms are discussed controversially, and remain to be identified in more detail”. I suggest changing to “However, there is controversy in the field regarding the molecular mechanism by which PS regulates autophagy”

Response: The sentence has been rephased according to the suggestion of this reviewer.

3. I could not find a  reference to Figure 1 in the text.

Response: We now refer to figure 1 in the revised manuscript

We would like to this reviewer for critical reading of the manuscript, very positive comments, and the very helpful comments and suggestions to improve the manuscript.